# Probiotics Treatment Improves Hippocampal Dependent Cognition in a Rodent Model of Parkinson’s Disease

**DOI:** 10.3390/microorganisms8111661

**Published:** 2020-10-27

**Authors:** Caroline Xie, Asheeta A. Prasad

**Affiliations:** School of Psychology, UNSW Sydney, Sydney NSW 2052, Australia; caroline.yl.xie@gmail.com

**Keywords:** Parkinson’s disease, probiotics, 6-Hydroxydopamine, non-motor symptoms, cognition, anxiety, rat, hippocampus

## Abstract

Parkinson’s disease (PD) is a neurological disorder with motor dysfunction and a number of psychiatric symptoms. Symptoms such as anxiety and cognitive deficits emerge prior to motor symptoms and persist over time. There are limited treatments targeting PD psychiatric symptoms. Emerging studies reveal that the gut microbe is altered in PD patients. Here we assessed the effect of a probiotic treatment in a rat model of PD. We used the neurotoxin (6-hydroxydopamine, 6-OHDA) in a preclinical PD model to examine the impact of a probiotic treatment (*Lacticaseibacillus rhamnosus* HA-114) on anxiety and memory. Rats underwent either sham surgery or received 6-OHDA bilaterally into the striatum. Three weeks post-surgery, rats were divided into three experimental groups: a sham group that received probiotics, a 6-OHDA group that received probiotics, and the third group of 6-OHDA received the placebo formula. All rats had access to either placebo or probiotics formula for 6 weeks. All groups were assessed for anxiety-like behaviour using the elevated plus maze. Cognition was assessed for both non-hippocampal and hippocampal dependent tasks using the novel object recognition and novel place recognition. We report that the 6-OHDA lesion induced anxiety-like behaviour and deficits in hippocampal dependent cognition. Interestingly, the probiotics treatment had no impact on anxiety-like behaviour but selectively improved hippocampal dependent cognition deficits. Together, the results presented here highlight the utility of animal models in examining the neuropsychiatric symptoms of PD and the potential of probiotics as adjunctive treatment for non-motor symptoms of PD.

## 1. Introduction

Parkinson’s disease (PD) is a neurodegenerative disorder, hallmarked by the loss of dopamine neurons in the substantia nigra (SN) [1]. PD patients suffer from a number of neuropsychiatric symptoms including cognitive impairment, sleep disorders, anxiety, and depression [2,3,4,5,6,7,8,9]. Some non-motor symptoms can precede years prior to the detection of early motor symptoms and continue with disease progression [5,8,10]. Anxiety and cognitive impairments are highly prevalent in PD populations, with 60% of PD patients reporting anxiety, 80% reporting cognitive deficits, and dementia affecting over 80% of PD patients [2,4,6,11,12,13]. Non-motor symptoms of PD often have a major impact on patient’s quality of life [8,13,14,15].

Post-mortem human brain analysis shows that individuals diagnosed with PD lose 60–90% of dopamine neurons in the SN [16,17]. Therefore, preclinical approaches for modelling PD include dopamine depletion using the neurotoxin 6-hydroxydopamine (6-OHDA). The 6-OHDA neurotoxin is taken up selectively by the dopamine transporter (DAT) leading to the release of free radicals within the cells, resulting in dopamine cell death [18,19]. To dissociate neuropsychiatric symptoms from motor deficits, it is crucial to deplete a subset of dopamine neurons, as direct administration of 6-OHDA into the SN either unilaterally or bilaterally leads to severe motor symptoms [20,21]. Induction of partial dopamine depletion by administering 6-OHDA into the striatum has been reliably applied to study neuropsychiatric symptoms of anxiety, cognitive deficits, and motivation [22,23,24,25,26,27,28,29,30,31]. 6-OHDA is taken up by DAT and expressed on nigral neurons projecting the striatum, hence, retrogradely depleting nigrostriatal neurons [24,30,32].

While there is no cure for PD, there are pharmacological treatments that replenish dopamine levels that are effective in reducing motor symptoms [1,33]. However, despite the high prevalence of neuropsychiatric symptoms in PD patients, there are limited treatments targeting neuropsychiatric symptoms of PD [4,9,34]. Emerging studies report significant changes in the microbiome composition in PD patients [35,36,37]. Significant changes in the gut microbiome show correlation with disease severity [35,36,37,38,39].

It is now evident that PD pathology includes an imbalanced gut microbiome [15,35,37,40,41,42,43]. Therefore, a treatment that “repairs” the microbiome might ameliorate some PD symptoms [44,45]. Several different probiotic treatments have shown to reduce anxiety-like behaviour and cognitive impairments in rodent models of neurological disorders [46,47,48,49]. In a preclinical model of Alzheimer’s disease, an 8-week treatment with probiotics improved spatial performance in the Morris water maze [49]. In a randomised, double-blind, and controlled trial with 60 Alzheimer’s patients, probiotic supplementation for 12 weeks showed significant improvement on cognitive function on the mini-mental state examination score [50]. Another probiotic formula (SLAB51), which includes nine different bacterial strains of bifidobacteria and lactobacilli, has also improved cognition and reduced amyloid plaque formation [51]. In the unilateral 6-OHDA model, SLAB51 probiotics were found to be neuroprotective from 6-OHDA toxicity and reduced inflammation [42].

Despite evidence of alterations in the microbiome of PD patients, and the findings that probiotics treatment alleviates neuropsychiatric symptoms, no studies to date have examined the potential of probiotics in treating the neuropsychiatric symptoms associated with PD. In this preclinical study, we used the 6-OHDA PD rat model to examine the impact of a probiotic treatment (*Lacticaseibacillus rhamnosus* HA-114). Probiotics mixtures consisting of lactobacilli administered to a variety of rodent models of neurological disorders have shown promising effects including reduction in PD motor dysfunction [42], progression of Alzheimer’s disease [51], and relapse to conditioned fear [52].

In this model of PD, three weeks post-surgery is sufficient to develop anxiety and cognitive symptoms [24,27,29,30,31]. Here we aimed to examine the therapeutic effect of probiotics after PD symptoms have developed. Hence after three weeks post-surgery, rats were given probiotics mixed in their drinking water in their home cages for six weeks. Administration of probiotics treatment for 5–10 weeks in rodent models of Alzheimer’s disease and PD has been deemed effective in rescuing symptoms [42,46,49].

After six weeks of treatment, assessments for anxiety-like behaviour and cognition (i.e., memory) commenced. The question of interest was whether treatment with probiotics can rescue deficits in anxiety and memory. To assess anxiety-like behaviour, we used the elevated plus maze (EPM), where anxious animals spend more time in the closed arms, relative to the open arms of the maze [53]. To assess for changes in cognition, we measured both hippocampal dependent and independent memory. To do this, animals were initially exposed to two novel objects in a large open field. For the hippocampal independent task, one of the objects were replaced at test (i.e., object recognition) while for the hippocampal dependent task, the same two objects were used but one was moved to a novel location (i.e., place recognition) [54,55]. The aim of the current study was to assess the impact of a specific bacterial strain (*Lacticaseibacillus rhamnosus* HA-114) on anxiety and cognition symptoms in the 6-OHDA rat model.

## 2. Materials and Methods

### 2.1. Subjects

Thirty-one experimentally naive adult male Sprague Dawley rats were obtained from the breeding colony in the School of Psychology at UNSW Sydney. Animals were between 3–4 months of age and weighed on an average 450 g at the time of surgery and were housed in groups of 3–4 per plastic box (60 cm × 30 cm × 12 cm; L × W × H) with wire top (total height 27.5 cm) and maintained on a 12 h light/dark cycle (lights on at 0700 h) with food and water available ad libitum. All procedures were conducted in accordance with the Australian Code of Practice for the Care and Use of Animals for Scientific Purposes (8th Edition, 2013) and approved (17/137B) by the UNSW Animal Care and Ethics Committee.

### 2.2. Surgery

Stereotaxic surgery was done as described previously [56]. Rats received intraperitoneal injections of a mixture of 1.3 mL/kg ketamine at a concentration of 100 mg/mL (Ketapex; Apex Laboratories, Sydney, Australia) and 0.3 mL/kg of muscle relaxant xylazine at a concentration 20 mg/kg (Rompun; Bayer, Sydney, Australia). Rats received a subcutaneous injection of 0.1 mL of a 5 mg/mL carprofen (Pfizer, Tadworth, United Kingdom) before being placed in the stereotaxic frame (Kopf Instruments, Tujunga, CA, USA). They received stereotaxic surgery using the flat skull coordinates according to Paxinos and Watson [57] atlas. Coordinates relative to bregma in millimetres were (+1.1 AP, ±3.2 ML, −7.2 DV), similar to coordinates used by [25] to infuse 6-OHDA to examine non-motor behaviours in a rat model of PD. The skull was exposed using a scalpel. Small holes at the coordinates were made using a hand drill. Rats then received bilateral infusions of either of 3 µL of saline or 6-OHDA (H116, Sigma Sydney, Australia). 6-OHDA was dissolved in 0.2% ascorbic acid to concentration of 4 µg/µL. A measure of 12–15 ug of 6-OHDA has previously been used to assess non-motor symptoms [25,32,35]. A total of 12 µg of 6-OHDA was infused per hemisphere into the striatum.

A Hamilton micro-syringe was lowered to the stereotaxic coordinates to deliver saline or 6-OHDA via 23-gauge, using a cone tipped 5 μL stainless steel injectors (SGE Analytical Science, Australia) at a rate of 1 μL/min using a UMP3 Micro-syringe Injector and Micro4 Controller (World Precision Instruments, Inc., Sarasota, FL, USA). The needle was left in place for 5 min to allow for diffusion. At the end of surgery, rats received an intramuscular injection of 0.2 mL of a 150 mg/mL solution of procaine penicillin (Benacillin; Troy Laboratories, NSW, Australia) and 0.2 mL of 100 mg/mL cephazolin sodium (AFT Pharmaceuticals, North Ryde, NSW, Australia). All rats were monitored for weight and behavioural changes post-surgery.

### 2.3. Probiotic Treatment

Three weeks post-surgery, rats were divided into three experimental groups: a sham group that received probiotics (n = 12), a 6-OHDA group that received probiotics (n = 10), and the third group of 6-OHDA that received the placebo formula (n = 9). A sample size of 8–10 rats per group was based on previous studies using probiotics to examine behavioural changes [49,51]. A probiotic strain consisting of *Lacticaseibacillus rhamnosus* HA-114 and a placebo formulation were both provided by Lallemand Health Solutions (Montreal, QC, Canada). Powdered probiotic was rehydrated with distilled water at a concentration of 109 colony forming units per millilitre (CFU) initially based on previous studies [55]. There was an observable increase of volume of liquid intake and increase in urine in the cages within the first couple of days. To decrease the volume consumed, the concentration was reduced 10^8^ CFU from the fifth day onwards. The placebo and probiotics mixtures were changed every second day to ensure bacteria viability. Rats received placebo or probiotics treatment for 6 weeks before behavioural tests commenced and continued to receive their respective treatment throughout the behavioural procedures.

### 2.4. Behavioural Procedures

A timeline of the general experimental procedures is described in Figure 1A. Details for the apparatus used in the behavioural tasks are provided in the Appendix A.

### 2.5. Novel Object Recognition (NOR) Task

In the NOR tasks, one object presented during the familiarisation session was now replaced with a second novel object. The objects used during the familiarisation session and test session were counter-balanced between animals to ensure the exploration of an object was not driven by preference for a particular object. After familiarisation, animals were returned to the arena after 5 min to test working memory retention. Animals were allowed to freely explore the arena for 3 min during the test session, and the time spent exploring each object was recorded. Following testing, the animal was returned to its home cage and both objects and the arena were cleaned with 70% ethanol.

### 2.6. Novel Place Recognition (NPR) Task

In the NPR task, after familiarisation to the arena, one of the objects presented during the familiarisation session was now placed in a novel location while the other remained in the same location as it was during the familiarisation phase. Animals were returned to the arena for testing after the 5 min to test short term memory retention. The original/novel locations were counterbalanced. Following a 3 min test session, the animal was returned to its home cage and both objects and the arena were cleaned with 70% ethanol.

### 2.7. Novelty Preference Ratio

For both the object and place recognition tasks, a retention ratio was derived for each animal, such that the time spent with the novel object (or the object in the novel location) at test was divided by the time spent with both objects (tnovel/tnovel + tfamiliar). Ratios of 0.5 indicate no memory (i.e., chance performance) while higher ratios indicate memory retention. If on any task, the animal explored either object for less than one second during the test phase, then familiarisation and test data from that task for that animal were excluded from the analysis.

### 2.8. Elevated Plus-Maze (EPM)

The elevated plus-maze (EPM) is a well-validated method for assessing anxiety in rodents, based on their general aversion to open spaces [53]. Rats were placed on the EPM apparatus facing the same open arm and allowed to freely explore for 5 min. A video camera placed above the apparatus recorded the test session and the amount of time spent in the open arm, and where all four legs were in the arm was measured. Data were recorded as the total amount of time spent in the open arm. A subset of the test data (minimum 4 rats per condition) was scored by a second observer who was blind to the experimental condition. Inter-observer reliability was high (correlation score = 0.88). Behavioural testing and scoring were carried out under blind conditions.

### 2.9. Immunohistochemistry

To confirm dopamine depletion, immunohistochemistry for anti-tyrosine hydroxylase (TH) was conducted using a similar protocol as described in Prasad and McNally, 2016 [56]. Briefly, sections were then incubated in sheep antiserum against TH (1:2000; Chemicon, #AB1542) in a PB solution containing 2% NHS and 0.2% Triton X-10 (48 h at 4 °C). The sections were then washed and incubated in biotinylated donkey anti-sheep (1:1000; 24 h at 4 °C; #713065147, Jackson ImmunoResearch Laboratories, West Grove, PA, USA). Microscopic images were captured using Olympus BX51 transmitted light microscope (Olympus, Tokyo, Japan). Manual counts for the number of TH-immunoreactive neurons were performed on Adobe Photoshop software. One section per brain within the SN mediolateral axis (bregma −5.20 to −5.55) was identified using Paxinos and Watson [57] atlas, and atlas was used to count the number of clearly delineated neuronal staining of tyrosine hydroxylase labelled neurons by an experimenter blinded by experimental group.

### 2.10. Statistical Analysis

All analyses were conducted using IBM SPSS Statistics Version 25. Analysis of Variance (ANOVA) was used to examine whether there were group differences and time spent in the open arms of the elevated plus maze across groups (sham-probiotics, PD-placebo, and PD-probiotics group). The Tukey post hoc test was used to further assess differences between groups. For the analysis of the object and place recognition procedures, the memory retention ratio of each group was compared to 0.5 (chance performance) using one sample *t*-tests [55,58]. If an animal had a novelty preference of 0 (i.e., it never encountered the novel object, or the familiar object in the novel location), then it was excluded from that particular analysis. Data in figures are expressed as the mean ± SEM. Statistical significance was set at *p* < 0.05.

## 3. Results

### 3.1. Weight and Formula Intake

There were three groups of rats: sham + probiotics (n = 12), PD + placebo (n = 9), and PD + probiotics (n = 10). Weight and formula intake were recorded during the 6-week period of treatment (Figure 1B,C). For the 6-week treatment period, the percentage of weight gain per group (mean ± SEM) were sham + probiotics (20.89 ± 3.59), PD + placebo (17.23 ± 0.88), and PD + probiotics (12.44 ± 1.48). There were no differences in percentage of weight gain, F (2, 30) = 2.77, *p* = 0.08. The amount of formula consumed was measured by weighing the drinking bottles [52]. The average intake of formula in a week per group were sham + probiotics (231.24 ± 25.81), PD + placebo (202.79 ± 14.77), and PD + Probiotics (251.09 ± 17.92). There were no differences amount of formula consumed, F (2, 30) = 1.59, *p* = 0.235.

### 3.2. Hippocampal Independent Cognition Remains Intact in 6-OHDA Lesioned Rats

The preference ratio for short term memory in the novel object recognition task for the groups were as follows: (mean ± SEM) for sham + probiotics n = 10, (0.738 ± 0.054); PD + placebo n = 8, (0.690 ± 0.072); and PD + probiotics n = 9, (0.778 ± 0.0427). Sham + probiotics group spent more time exploring the novel object (t_10_ = 4.39, *p* = 0.001), indicating good memory retention. Similarly, both the PD groups also exhibited good memory retention on this task indicated by higher preference to explore the novel object than the familiar object (PD + placebo, t_8_ = 2.60, *p* = 0.031, and PD + probiotics, t_9_ = 6.52, *p* < 0.001). Hence, no memory impairment was induced by the dopamine depletion when animals were tested on the non-hippocampal object recognition task after a 5 min retention interval, Figure 2A.

### 3.3. Hippocampal Dependent Cognitive Deficits in 6-OHDA Lesioned Rats Are Reversed by Probiotics

The preference ratio for short term memory in the novel place recognition task for the groups were as follows: (mean ± SEM) for sham + probiotics n = 10, (0.675 ± 0.047); PD + placebo n = 8, (0.575 ± 0.074); and PD + probiotics n = 8, (0.705 ± 0.079). The sham group spent more time exploring the familiar object that had been moved to a novel location (t_10_ = 3.71, *p* = 0.004), indicating good retention. However, the PD + placebo group were impaired in this task and exhibited no evidence of memory (t_8_ = 0.51, *p* = 0.62). In other words, there was a memory impairment induced by the dopamine depletion when animals were tested on the hippocampal dependent place recognition task. Remarkably, this deficit was rescued in the PD + probiotics group as they exhibited good retention (t_8_ = 2.59, *p* = 0.032), Figure 2B.

### 3.4. Probiotics Treatment Does Not Impact Anxiety Behaviour in 6-OHDA-Lesioned Rats

Elevated plus maze (EPM) is a simple yet effective method for assessing anxiety related behaviour in rodents [53,59]. Replicating past studies of the 6-OHDA model of PD, we found an increase in anxiety-like behaviour in dopamine lesioned animals when tested on the EPM [23,60]. All rats were included in the analysis: sham group (n = 12), PD + placebo (n = 9), and PD + probiotics (n = 10). There was significant difference in time spent in open arms between the sham group and PD groups in seconds (mean ± SEM): sham (46.58 ± 9.41), PD + placebo (13.44 ±3.64), and PD + probiotics (11.20 ± 3.11). That is, the sham group spent significantly more time on the open arms than the PD groups, F (2,30) = 8.995, *p* = 0.001. However, there was no difference between PD + placebo and PD + probiotics group, *p* = 0.973, Figure 2C.

### 3.5. Selective Dopamine Depletion in the Substantia Nigra

To verify the extent of dopamine loss, the number the dopamine neurons within the SN were counted. During the immunohistochemical process, one sample from the sham group got fragmented possibly due to insufficient fixation during perfusion, thus excluded from analysis, so sham + probiotics (n = 11), PD + placebo (n = 9), and PD + probiotics (n = 10). Quantification of immuno-reactivity was carried out under blind conditions. Cell counts for immuno-reactivity for dopamine revealed a significant decrease of dopamine neurons in the 6-OHDA infused groups (mean ± SEM), PD + placebo (62.77 ± 7.51), and PD + probiotics (72.88 ± 4.93), relative to the sham + probiotics (121.00 ± 15.19), F (2, 29) = 8.353, *p* = 0.001. There was no difference in number of dopamine neurons between the two PD groups (i.e., PD + placebo and the PD + probiotics group, *p* = 0.801), as shown in Figure 2D. Representative TH-immunoreactive images at high magnification shown in Figure 2E showed similar levels of dopamine depletion in the two PD groups.

## 4. Discussion

The neurobiology of the non-motor symptoms in PD is poorly understood. The responsiveness of dopamine therapy on some non-motor symptoms are indicative of dopamine links [2]. Depletion of nigrostriatal dopamine neurons in standard rodent model was used to examine neuropsychiatric symptoms of PD including cognitive deficits, motivation, and anxiety [23,24,27,60,61,62]. There were no changes in weight gain and formula intake across groups indicating other functions were intact. Consistent with previous studies using EPM to assess anxiety, the dopamine lesion clearly induced anxiety-like behaviour [24,27,29,30,31]. There was a significant reduction in exploration of the open arms of the EPM in the dopamine-depleted rats compared to the sham group, showing dopamine depletion increases anxiety. However, no difference in EPM performance was detected between the PD + placebo and PD + probiotic group, suggesting no significant impact of probiotic treatment on anxiety behaviour. There is a spectrum of anxiety symptoms in PD patients, where in some PD cases the “wearing off” of levodopa is correlated with anxiety [63]. Our results show dopamine dependent anxiety, and that probiotics treatment was not effective in alleviating dopamine related anxiety.

The effectiveness of EPM is fundamentally based on the innate aversion of rodents to open spaces and elevation [59]. Other measures for anxiety behaviours include observations in an open field [63,64]. An anxious animal will spend more time on the outer parameter of the open field compared to the time spent in the inner zone, showing a reduced exploratory behaviour [65,66]. Assessment of anxiety-like behaviour in the open field would be an additional measure to clearly demonstrate the anxiety detected in the PD groups. Nonetheless, in addition to EPM, we also assessed cognition in an open arena. For both cognition assays, the animal explores objects in the centre of the open arena. All animals in our study explored at least one object at the centre of arena, indicating all rats in this study exhibited exploratory behaviour. Specifically, for the object recognition task where the animals had to distinguish between two different objects placed in the centre of the arena, there were no differences between all groups.

The impact on cognitive function in the dopamine depletion (6-OHDA model) has mainly been tested using the Morris water maze [24,27]. The Morris water maze is a classical test for assessing changes in spatial memory. It is useful to assess both acquisition and retention of memory [67]. In the present study, we used other cognition assessments of novel place and novel object recognition test. Both these tasks exploit the natural inclination of rodents to explore novelty [55]. These two tests can be applied to distinguish between hippocampus dependent and independent cognition. In the non-hippocampal dependent assay, object recognition test recruits the perirhinal cortex [68]. In our study, all groups displayed good memory retention in the non-hippocampal task.

For the hippocampal dependent task, the sham group displayed good retention for place recognition. Interestingly, there was a specific deficit in the PD + placebo group during this spatial task. Remarkably, this deficit was rescued by probiotics treatment. Using the Morris water maze to assess hippocampal dependent cognition, the 6-OHDA rat model showed latency to reach the cued platform [24]. Together with previous studies using the Morris water maze, our study also shows hippocampal dependent cognitive deficits.

The outcomes from this study show two key advances in knowledge. Firstly, the utility of animal models to assess neuropsychiatric symptoms and secondly, the potential of probiotics as an adjunctive treatment for PD.

The dopamine lesion specifically had no impact on non-hippocampal dependent task, with all groups showing comparable memory retention for object recognition task. Yet, dopamine lesion selectively impaired the hippocampal dependent task, showing that dopamine activity is essential for spatial cognition. Similarly, PD patients assessed on and off dopaminergic treatment using virtual reality spatial navigation task showed dopaminergic treatment improved spatial navigation [69]. In animal models, administration of D1/D5 receptor antagonist into the hippocampus blocks spatial memory performance [70]. PD patients show greater cognitive impairment in spatial working memory compared to object recognition memory [71,72]. This selective cognitive impairment in our 6-OHDA PD model is consistent with specific impairments in PD patients, reflecting on the utility of using animal models in examining PD neuropsychiatric symptoms. Interestingly, magnetic resonance imaging (MRI) scans show PD patients have smaller volumes of total hippocampus compared to normal controls. Moreover, there were no significant difference perirhinal cortex size [73]. Hence in our rodent study, dopamine depletion may not have affected the perirhinal cortex; therefore, no deficits on non-hippocampal dependent cognitive changes could be detected and nor could the impact of probiotics be assessed here. Supplementary treatment of probiotics for PD patients may serve to ameliorate the hippocampal dependent cognitive deficits.

The impact of probiotics on non-motor symptoms suggest changes in the gut-brain axis [44,74]. Analogous to the correlation of motor severity and changes in the microbiome of PD patients, there may be specific microbe changes associated with non-motor symptoms of PD patients [35,37]. Probiotics have been successfully applied in Alzheimer’s preclinical and clinical studies, showing improvement in cognitive function with the potential to reduce oxidative stress, metabolic state, and inflammatory processes [49,50,51]. Here, for the first time, we report improvement in cognitive function after probiotics treatment in a PD rat model. However, the underlying mechanism for the effectiveness of probiotics remains outstanding in this study. Probiotics have diverse functions including modulation of inflammation, neurotransmitters, enzymatic activities, and neurotrophic molecules [75]. In the unilateral 6-OHDA model, administration of probiotics (SLAB5) formula given two weeks prior to dopamine lesion increased neuroprotection of dopamine neurons and decreased motor dysfunction [42]. It is possible that the inflammatory modulation mechanism of probiotics is similar in non-motor symptoms. As gut microbiome has been shown to regulate inflammatory processes in the ventral hippocampus [76]. Other rodent studies have shown that increase in inflammation and oxidative stress impairs spatial cognition [54]. We suspect similar mechanism of anti-inflammatory properties underly the impact of *Lacticaseibacillus rhamnosus* HA-114 on cognitive function. The probiotic SLAB5 is a mixture of two bacterial strains: bifidobacteria and lactobacilli [45]. It is possible that specific bacterial strains have diverse functions including differ in. However, further research is required to elucidate the mechanisms of action of probiotics in PD non-motor symptoms.

## 5. Conclusions

Our data show that probiotic treatment rescues hippocampal dependent deficits. Overall, the results presented here highlight the utility of animal models in examining neuropsychiatric symptoms of PD and the potential of probiotics as a supplementary treatment for PD.

## Figures and Tables

**Figure 1 microorganisms-08-01661-f001:**
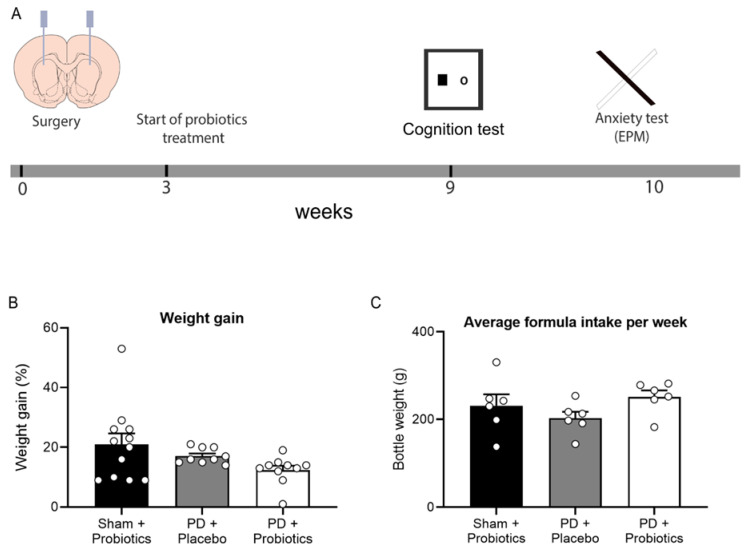
(**A**) Timeline of experimental procedures. Rats received intracranial surgery to deliver saline or 6-OHDA into the striatum. Three weeks post-surgery, placebo/probiotics formula was given to rats in their home cages. After six weeks of formula treatment (9 weeks post-surgery), behavioural assessments began. (**B**) Graph of percentage of weight gain during 6 weeks of formula treatment. No group differences between groups, *p* = 0.08. (**C**) Graph of average formula intake per week during 6 weeks of formula treatment, no group differences between groups, *p* = 0.235.

**Figure 2 microorganisms-08-01661-f002:**
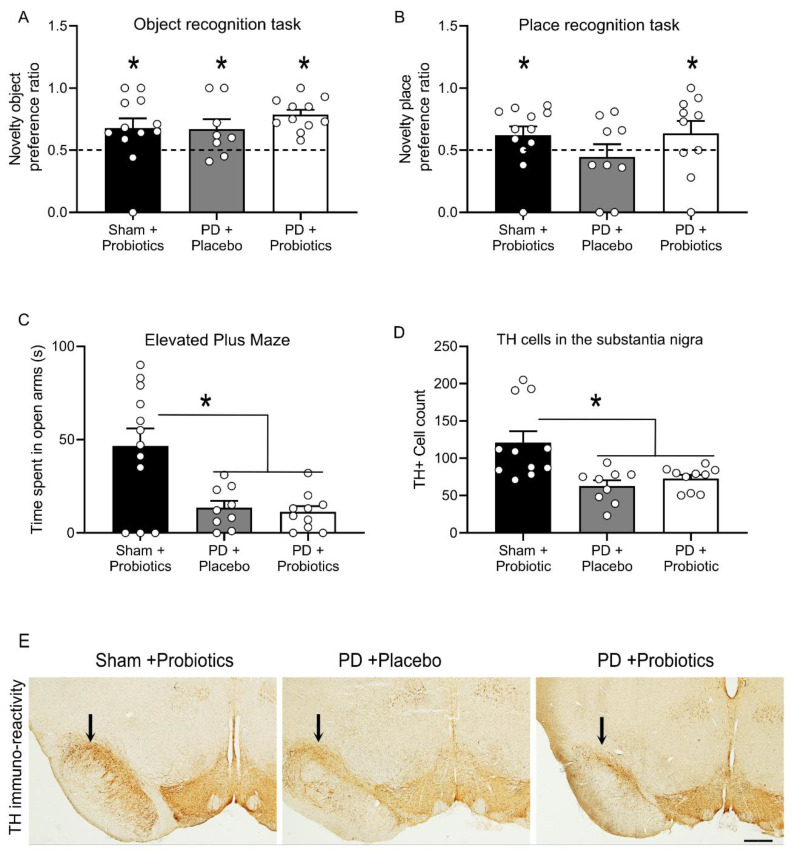
Performance on the novel object and novel place tasks, where the novelty preference ratio for the object and place tasks are depicted in panels (**A**,**B**) (respectively). “*” indicates retention ratios larger than 0.5, demonstrating memory retention of the familiar object/place, indicated by the dashed line set at 0.5. (**C**) Time, in seconds, spent in the open arms of the elevated plus maze (mean ± SEM). There was significant difference between sham and Parkinson’s disease (PD) groups, indicating greater anxiety in the PD groups; the “*” indicates *p* = 0.001. There was no difference between the two PD groups, *p* = 0.973. (**D**) Graph showing the tyrosine hydroxylase (TH) immunoreactive neuronal count between groups. There were significantly less dopamine neurons in the PD groups compared to sham group. “*” indicates *p* = 0.001. There was no difference between the two PD groups, *p* = 0.801. (**E**) Representative images of immunohistochemical stain of tyrosine hydroxylase in the substantia nigra. Arrows above substantia nigra indicate region of dopamine loss. Images taken 10×, scale bar = 100 microns.

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
