# Peer review of "Probiotics Treatment Improves Hippocampal Dependent Cognition in a Rodent Model of Parkinson’s Disease"

_microorganisms, 2020, doi:10.3390/microorganisms8111661_

Round 1
Reviewer 1 Report
The Ms by Xie and Prasad addresses the question as to whether the probiotic (Lacticaseibacillus Rhamnosus H114) supplementation for 6 weeks affects behavioural disorders (anxiety and cognition deficits) reported in rats bearing a monolateral striatal lesion induced by focal microinjection of 6-OHDA in the striatum. The results show that, in 6-OHDA lesioned rats the latter supplementation has no significant effect on: 1) neurpathological damage (TH positive neuronal count); 2) anxiety; 3) hippocampal dependent cognition. At variance with the above negative results, 6-OHDA induced deficit in hippocampal independent cognition is minimized by probiotic supplementation.
The title reflects the content of the Ms and the abstract is informative. The introductory paragraph sets well the scenario under which the research project has been undertaken. The methodologies selected are well described, appropriate to the scope and used competently by the Auhtors. The results generated are described consistently with the current literature.
This referee has no major criticisms to make to the rationale of the work. However, the experimental protocol lacks of some important information that undermine the impact of the study. For instance, 1) it is not stated how the Authors have calculated the number of animal per group to be used (i.e. sample power calculation); 2) it is not said whether the treatment schedule, immunoistochemistry procedures and behavioural assessment were carried out under blind conditions; 3) the Authors have selected a rat of 450 g for their PD model with no explanation or comparative analysis with similar model in terms of neuropathological and behavioural response. These questions are important per se (i.e. the 3R approach to aninaml studies is not satisfied) but also in the light of the lack of neuroprotection reported here that appears at variance with data of the literature. These aspect may also form reason for deepening the discussion in terms of mechanism of probiotic action. For instance: colonization by the bacterial species used here is directly related to the lack of neuroprotection and minimization of hippocampal independent cognition? Alternatively, could the results (negative and positive) reported here stem from alteration of the proportion of other species in the gut (e.g. Enterobacteriaceae and Enterococcaceae)? Despite these latter questions need an experimental answer the Authors may attempt some critical analysis of their data and those present in the literature.
Author Response
The Ms by Xie and Prasad addresses the question as to whether the probiotic (Lacticaseibacillus Rhamnosus H114) supplementation for 6 weeks affects behavioural disorders (anxiety and cognition deficits) reported in rats bearing a monolateral striatal lesion induced by focal microinjection of 6-OHDA in the striatum. The results show that, in 6-OHDA lesioned rats the latter supplementation has no significant effect on: 1) neurpathological damage (TH positive neuronal count); 2) anxiety; 3) hippocampal dependent cognition. At variance with the above negative results, 6-OHDA induced deficit in hippocampal independent cognition is minimized by probiotic supplementation.
The title reflects the content of the Ms and the abstract is informative. The introductory paragraph sets well the scenario under which the research project has been undertaken. The methodologies selected are well described, appropriate to the scope and used competently by the Auhtors. The results generated are described consistently with the current literature.
This referee has no major criticisms to make to the rationale of the work. However, the experimental protocol lacks of some important information that undermine the impact of the study.
For instance, 1) it is not stated how the Authors have calculated the number of animal per group to be used (i.e. sample power calculation)
“Sample size of 8-10 rats per group was based on previous studies using probiotics to examine behavioral changes (Bonfili et al., 2017; Asl et al., 2019” is now added to the methods section
2) it is not said whether the treatment schedule, immunoistochemistry procedures and behavioural assessment were carried out under blind conditions;
“Behavioral testing and scoring were carried out under blind conditions” added in methods section
“Quantification of immuno-reactivity was carried out under blind conditions” added in results section
3) the Authors have selected a rat of 450 g for their PD model with no explanation or comparative analysis with similar model in terms of neuropathological and behavioural response. These questions are important per se (i.e. the 3R approach to aninaml studies is not satisfied) but also in the light of the lack of neuroprotection reported here that appears at variance with data of the literature. These aspect may also form reason for deepening the discussion in terms of mechanism of probiotic action. For instance: colonization by the bacterial species used here is directly related to the lack of neuroprotection and minimization of hippocampal independent cognition? Alternatively, could the results (negative and positive) reported here stem from alteration of the proportion of other species in the gut (e.g. Enterobacteriaceae and Enterococcaceae)? Despite these latter questions need an experimental answer the Authors may attempt some critical analysis of their data and those present in the literature.
The animals were between 3-4 months of age and weighed on an average 450g at the time of the surgery. Generally adults are used in PD rodent models. All groups were weighed weekly and gained the similar amounts of weight during the experimental procedure.
Thank you for the suggestion to deepen the discussion with possible mechanism of probiotic action.
The discussion now includes “The probiotic SLAB5 is a mixture of two bacterial strains; bifidobacteria and lactobacilli (Castelli et al., 2020). It is possible that specific bacterial strain has distinct function such as bifidobacteria impact neuroprotection and lactobacilli influences behaviour.”

Reviewer 2 Report
In this manuscript, Xie & Prasad attempt to validate 6-OHDA induced PD model to study the effect of probiotic consumption on cognitive and anxiety behavior in PD. This is an interesting hypothesis but is poorly developed in this manuscript. Although probiotic consumption did not increase TH+ neurons in SN, the authors found that it improved hippocampus-dependent memory in the animal model of PD.
Major
Missing locomotion data- to validate the injection in striatum did not produce locomotor deficits in the cohort used in this study. Especially because authors describe in detail in the introduction the selection of striatum over SN. Although authors cannot quantify distance traveled due to technical limitations, they must quantify the total mobile time in each test and show whether the cohort used in the study developed any motor symptoms or not.
Many references are missing from the reference list; here are a few examples. The authors must cross-check the reference list for cited literature.
Line 141: Prasad et al 2016 reference is missing in the reference list.
Line 50: Abbott et al 2005 reference is missing in the reference list.
Line 54: Hely et al 2008 reference is missing in the reference list.
Line 55: Schrag et al 2000 reference is missing in the reference list.
Line 59: Damier et al 1999 reference is missing in the reference list.
All the bar graphs must be replaced with the scatter dot plot that shows the distribution of data.
The authors must replace all the figures with high-resolution figures.
Line 105-106: Why was this particular probiotic bacteria selected to study? The rationale for selecting this particular bacteria is missing in the manuscript.
Line 153-154: Why was the particular dose of 12ug selected? Provide the reference and rationale for this dose
The authors must re-arrange the sequence of figures and results to match the time course of study. E.g. TH immunoreactivity was determined at the end of the behavioral study so it must be the last figure and must be described at the end of the results section. Likewise, EPM was performed after memory tests but is described before the memory test in the results section.
Similarly, the Discussion section should also follow the timeline of study.
The relevance of weight gain and formula intake results must be discussed in the Discussion section.
Minor
There are many typographical errors e.g. line 19: overtime instead of overtime Line 66: serve instead of severe. The author must proofread the manuscript.
Line 131: The author must provide the specific age of adult rats used in this study.
Figure 1C: what does the y-axis label represent? Weight of bottle or weight of the liquid consumed.
Figure 1D & 1E are redundant. The author should keep only one panel and move the other panel to supplementary material.
Line 384-385: This sentence must be restructured.
Line 405: Mistitled as Conclusion. Change the title to Discussion.
References style must be numbered as specified by the journal.
Author Response
Major
Missing locomotion data- to validate the injection in striatum did not produce locomotor deficits in the cohort used in this study. Especially because authors describe in detail in the introduction the selection of striatum over SN. Although authors cannot quantify distance traveled due to technical limitations, they must quantify the total mobile time in each test and show whether the cohort used in the study developed any motor symptoms or not.
In the behaviors tested, immobility cannot be appropriately quantified. During the EPM test even though the PD animals spent more time in the closed arms; they were mobile as exploratory behavior is required to distinguish between the open and closed arms. In the cognition tests which require the animals to move to the novel objects or places, we report no difference between PD groups explored novel object at similar level as sham group.
Morris water maze or forced swim test where immobility is clearly detected would have been good assays to assess mobile time.
Many references are missing from the reference list; here are a few examples. The authors must cross-check the reference list for cited literature.
Line 141: Prasad et al 2016 reference is missing in the reference list.
Line 50: Abbott et al 2005 reference is missing in the reference list.
Line 54: Hely et al 2008 reference is missing in the reference list.
Line 55: Schrag et al 2000 reference is missing in the reference list.
Line 59: Damier et al 1999 reference is missing in the reference list.
These references are now added to the reference list.
Damier, P., Hirsch, E. C., Agid, Y., & Graybiel, A. M. (1999). The substantia nigra of the human brain: II. Patterns of loss of dopamine-containing neurons in Parkinson's disease. Brain, 122(8), 1437-1448. doi:10.1093/brain/122.8.1437
Hely, M. A., Reid, W. G. J., Adena, M. A., Halliday, G. M., & Morris, J. G. L. (2008). The Sydney multicenter study of Parkinson's disease: The inevitability of dementia at 20 years. Movement Disorders, 23(6), 837-844. doi:10.1002/mds.21956
Prasad, A. A., & McNally, G. P. (2016). Ventral Pallidum Output Pathways in Context-Induced Reinstatement of Alcohol Seeking. J Neurosci, 36(46), 11716-11726. doi:10.1523/jneurosci.2580-16.2016
Schrag, A., Jahanshahi, M., & Quinn, N. (2000). How does Parkinson's disease affect quality of life? A comparison with quality of life in the general population. Mov Disord, 15(6), 1112-1118. doi:10.1002/1531-8257(200011)15:6<1112::aid-mds1008>3.0.co;2-a
All the bar graphs must be replaced with the scatter dot plot that shows the distribution of data.
All graphs now show individual data points
The authors must replace all the figures with high-resolution figures.
All figures are replaced with high-resolution figures at the end of the manuscript
Line 105-106: Why was this particular probiotic bacteria selected to study? The rationale for selecting this particular bacteria is missing in the manuscript.
Our rationale for selecting the bacterial strain was from Cowan et al., 2016 where the authors found probiotics mixture of Lactobacillus rhamnosus and L. helveticus provided by Lallemand Health Solutions reduced relapse to fear related behaviour.
Upon contacting the supplier, they had some preliminary results indicating that a related bacterial strain (Lacticaseibacillus rhamnosus HA-114) was effective in non-motor symptoms in other neurodegenerative disorder. Those results are not published yet. In literature search we found lactobacilli was effective in neurological disorders.
We have added the below to the manuscript.
“Probiotics mixtures consisting lactobacilli administered to a variety of rodent models of neurological disorders has shown promising effects including reduction in PD motor dysfunction (Castelli et al., 2020), progression of Alzheimer’s disease (Bonfili et al., 2017) and relapse to conditioned fear (Cowan et al., 2016)”
Line 153-154: Why was the particular dose of 12ug selected? Provide the reference and rationale for this dose
“12-15 ug of 6-OHDA has previously used to assess non-motor symptoms (Tadaiesky et al., 2008; Chen et al., 2014; Magnard et al., 2016).” Added to methods section
The authors must re-arrange the sequence of figures and results to match the time course of study. E.g. TH immunoreactivity was determined at the end of the behavioral study so it must be the last figure and must be described at the end of the results section. Likewise, EPM was performed after memory tests but is described before the memory test in the results section.
Sequence of figures are re-arranged to match the time course of study, figure 1D removed. Similarly, the Discussion section should also follow the timeline of study.
Figures and legends are at the end of the manuscript
The relevance of weight gain and formula intake results must be discussed in the Discussion section.
“There were no changes in weight gain and formula intake across groups indicating other functions were intact” added in discussion
Minor
There are many typographical errors e.g. line 19: overtime instead of overtime Line 66: serve instead of severe. The author must proofread the manuscript.
Overtime changed to over time.
Serve changed to severe
Line 131: The author must provide the specific age of adult rats used in this study.
“Animals were between 3-4 months of age” added
Figure 1C: what does the y-axis label represent? Weight of bottle or weight of the liquid consumed.
‘The amount of formula consumed was measured by the weighing the drinking bottles” is the result section to describe the y axis.
Figure 1D & 1E are redundant. The author should keep only one panel and move the other panel to supplementary material.
Removed Figure 1D
Line 384-385: This sentence must be restructured.
Revised to “Hippocampal dependent cognitive deficits in 6-OHDA lesioned rats are reversed by probiotics.”
From "Cognitive deficits in short term memory the hippocampal dependent (novel place recognition) in 6-OHDA lesioned rats are reversed by probiotics."
Line 405: Mistitled as Conclusion. Change the title to Discussion.
Conclusion title is now changed to Discussion title
References style must be numbered as specified by the journal.
Reference styles is now numbered

Reviewer 3 Report
The manuscript absolutely does not meet the requirements of the Microorganisms. Authors should revised it using Microsoft Word template or LaTeX template
The abstract is overloaded with details and needs to be revised.
Figures 1 and 2 should be revised and presented in good quality.
In terms of experimental design, the lack of a placebo sharm group seems unfounded.
Line 92-97 The citation of articles describing the results of probiotic use in Alzheimer's disease in thismanuscript is not justified, since the mechanisms of Alzheimer's disease and Parkinson's disease have significant differences. Or the rationale for this citation should be more explicit.
Line 138 The number of UNSW Animal Care and Ethics Committee decision should be added.
The results should be discussed in the Discussion section, and the Conclusions should contain a short summary of the investigation.
Microorganisms are primarily focused on microorganisms, while the manuscript details the damage in Parkinson's disease, but ignores the probiotics used in the study. Authors should presented more information about Lacticaseibacillus rhamnosus HA-114 strain in Introduction as well as Methods. The authors should confirm that probiotics taken by animals with drinking water may indeed somehow influence the status of animals with Parkinson's disease. Are they not damaged in the stomach of animals? In what part of the digestive tract do they function?
Author Response
The manuscript absolutely does not meet the requirements of the Microorganisms. Authors should revised it using Microsoft Word template or LaTeX template
The manuscript has been revised using Microsoft Word template
The abstract is overloaded with details and needs to be revised.
The word count is less than 250-word limit requirements of the Microorganisms and contains details relevant for selective readership of Microorganisms
Figures 1 and 2 should be revised and presented in good quality.
Figures have been revised and improved in quality. Figures and legends are at the end of the manuscript.
In terms of experimental design, the lack of a placebo sharm group seems unfounded.
When designing the experiments we considered adding an additional placebo sham as you suggest. However, we would expect the intact control to display low level of anxiety, similar to what we found in the sham group. Sham group displayed good object and place cognition. If anything, the intact control group would display better performance in all behavioral assessments, yielding a larger difference when compared to the PD groups. Considering the “Reduction” framework for humane animal research we would obtain comparable levels of information from the intact control group.
Line 92-97 The citation of articles describing the results of probiotic use in Alzheimer's disease in this manuscript is not justified, since the mechanisms of Alzheimer's disease and Parkinson's disease have significant differences. Or the rationale for this citation should be more explicit.
“Probiotics mixtures consisting lactobacilli administered to a variety of rodent models of neurological disorders has shown promising effects including reduction in PD motor dysfunction (Castelli et al., 2020) the progression of Alzheimer’s disease (Bonfili et al., 2017) and relapse to conditioned fear (Cowan et al., 2016)”
Line 138 The number of UNSW Animal Care and Ethics Committee decision should be added.
Ethics approval number is added “17/137B”
The results should be discussed in the Discussion section, and the Conclusions should contain a short summary of the investigation.
Conclusion title is now changed to Discussion title
Microorganisms are primarily focused on microorganisms, while the manuscript details the damage in Parkinson's disease, but ignores the probiotics used in the study. Authors should presented more information about Lacticaseibacillus rhamnosus HA-114 strain in Introduction as well as Methods. The authors should confirm that probiotics taken by animals with drinking water may indeed somehow influence the status of animals with Parkinson's disease. Are they not damaged in the stomach of animals? In what part of the digestive tract do they function?
We have tried to include more information about the probiotics throughout the manuscript
We do show probiotics taken by animals with drinking water indeed influences the animals with Parkinson's disease spatial cognitive function. We do not have access to facilities yet to examine change in microbiological status in animals with Parkinson's disease.
There is considerable evidence that gut microbiome alters in PD patients. The fascinating area of research we are interested in is how the gut microbiome is altered when dopamine is lost. This manuscript will be the stimulus to expand collaboration with experts in microbiology to examine changes brain impact the gut microbiome. We are hopeful that this publication will generate interest in the readership of Microorganisms to share skills in neurological disorders.
Are they not damaged in the stomach of animals? In what part of the digestive tract do they function?
Microbiomes are present throughout the digestive tract; the specific localisation of function would be challenging to identify. Unilateral 6-OHDA has recently reported to induce colonic inflammation and gut mucosal barrier impairment (Pellegrinia et al., 2020).

Round 2
Reviewer 2 Report
The authors have addressed all the comments except the one major comment. They did not show the total mobile time (i.e. moving not stationary) in any of the tests. This is essential to validate the cohort used in this study and can be easily calculated from the videos. Authors reply that PD group animals spent more time in the closed arm does not distinguish whether the animals were moving or stationary in the arm. Novelty preference ratio is independent of total mobility because when animals are not exploring the objects or place, they are exploring the remaining part of the test arena.
line 247: Non-hippocampal dependent is an incorrect term. Please correct it to Hippocampal independent.
Author Response
The authors have addressed all the comments except the one major comment. They did not show the total mobile time (i.e. moving not stationary) in any of the tests. This is essential to validate the cohort used in this study and can be easily calculated from the videos. Authors reply that PD group animals spent more time in the closed arm does not distinguish whether the animals were moving or stationary in the arm. Novelty preference ratio is independent of total mobility because when animals are not exploring the objects or place, they are exploring the remaining part of the test arena.
We analysed the mobile time during the elevated plus maze (5 min test session). The data is now included the supplementary figure 1A and 1B showing time mobile and the number of entries into the open, closed and center zone during the test session.
line 247: Non-hippocampal dependent is an incorrect term. Please correct it to Hippocampal independent.
Corrected and reflected in line 225 (due to shift to Microorganism template)
Reviewer 3 Report
In fact, the authors did not take into account all the recommendations of the reviewer, which did not make the manuscript publicly available.
The list of references and reference numbers in the text are not formatted properly.
(See Instructions for authors:
In the text, reference numbers should be placed in square brackets [ ]
Author 1, A.B.; Author 2, C.D. Title of the article. Abbreviated Journal Name Year, Volume, page range.)
Figures are presented in very bad quality. I did not see any difference between the new and old versions and there are no new figures at the end of the manuscript, as the authors replied. The quality of Figure 2 does not allow me to see the differences between data and images.
The authors entered the necessary information about experimental procedures into the Methods, however, the reasons why this strain of probiotics was selected for the study remained unclear. The cited articles (in Introduction) refer to other strains. It is not clear if this strain has been studied as a probiotic before. The authors pay great attention to the processes occurring in Parkinson's disease, but for the journal Microorganisms, no less attention is required to pay the strain of the microorganism used in the study.
Author Response
In fact, the authors did not take into account all the recommendations of the reviewer, which did not make the manuscript publicly available.
The list of references and reference numbers in the text are not formatted properly.
(See Instructions for authors:
In the text, reference numbers should be placed in square brackets [1-10]
Author 1, A.B.; Author 2, C.D. Title of the article. Abbreviated Journal Name Year, Volume, page range.)
The manuscript is now revised using the Microorganism template
Figures are presented in very bad quality. I did not see any difference between the new and old versions and there are no new figures at the end of the manuscript, as the authors replied. The quality of Figure 2 does not allow me to see the differences between data and images.
Figure 2 has been revised. Apologies for referring to the figures at the end, I assumed the figures when uploaded separately would appear at the end of the file.
The authors entered the necessary information about experimental procedures into the Methods, however, the reasons why this strain of probiotics was selected for the study remained unclear. The cited articles (in Introduction) refer to other strains. It is not clear if this strain has been studied as a probiotic before. The authors pay great attention to the processes occurring in Parkinson's disease, but for the journal Microorganisms, no less attention is required to pay the strain of the microorganism used in the study.
Our rationale for selecting the bacterial strain was from Cowan et al., 2016 where the authors found probiotics mixture of Lactobacillus rhamnosus and L. helveticus provided by Lallemand Health Solutions reduced relapse to fear related behaviour.
Upon contacting the supplier, they had some preliminary results indicating that a related bacterial strain (Lacticaseibacillus rhamnosus HA-114) was effective in non-motor symptoms in other neurodegenerative disorder. Those results are not published yet. In literature search we found lactobacilli was effective in neurological disorders.
We added the below to the manuscript in revision 1
“Probiotics mixtures consisting lactobacilli administered to a variety of rodent models of neurological disorders has shown promising effects including reduction in PD motor dysfunction [45] progression of Alzheimer’s disease [54]and relapse to conditioned fear [55].”